# Cancer Stem Cells in Head and Neck Metastatic Malignant Melanoma Express Components of the Renin-Angiotensin System

**DOI:** 10.3390/life10110268

**Published:** 2020-11-02

**Authors:** Sam Siljee, Tessa Pilkington, Helen D. Brasch, Nicholas Bockett, Josie Patel, Erin Paterson, Paul F. Davis, Swee T. Tan

**Affiliations:** 1Gillies McIndoe Research Institute, Wellington 6021, New Zealand; sam.siljee@gmri.org.nz (S.S.); tessamp064@gmail.com (T.P.); helen.brasch@gmri.org.nz (H.D.B.); nick.bockett@gmri.org.nz (N.B.); josie.patel@gmri.org.nz (J.P.); erin.paterson@gmri.org.nz (E.P.); paul.davis@gmri.org.nz (P.F.D.); 2Wellington Regional Plastic, Maxillofacial and Burns Unit, Hutt Hospital, Wellington 5010, New Zealand; 3Department of Surgery, The Royal Melbourne Hospital, The University of Melbourne, Parkville, Victoria 3050, Australia

**Keywords:** metastatic malignant melanoma, head and neck, cancer stem cells, renin-angiotensin system, renin, pro-renin receptor, angiotensin-converting enzyme, angiotensin-converting enzyme 2, angiotensin II receptor 1, angiotensin II receptor 2

## Abstract

Components of the renin-angiotensin system (RAS) are expressed by cancer stem cells (CSCs) in many cancer types. We here investigated expression of the RAS by the CSC subpopulations in human head and neck metastatic malignant melanoma (HNmMM) tissue samples and HNmMM-derived primary cell lines. Immunohistochemical staining demonstrated expression of pro-renin receptor (PRR), angiotensin-converting enzyme (ACE), and angiotensin II receptor 2 (AT_2_R) in all; renin in one; and ACE2 in none of the 20 HNmMM tissue samples. PRR was localized to cells within the tumor nests (TNs), while AT_2_R was expressed by cells within the TNs and the peritumoral stroma (PTS). ACE was localized to the endothelium of the tumor microvessels within the PTS. Reverse transcription quantitative polymerase chain reaction (RT-qPCR) detected transcripts for PRR, ACE, ACE2, and AT_1_R, in all the five HNmMM tissue samples and four HNmMM-derived primary cell lines; renin in one tissue sample and one cell line, and AT_2_R in none of the five HNmMM tissue samples and cell lines. Western blotting showed variable expression of ACE, PRR, and AT_2_R, but not ACE2, in six HNmMM tissue samples and two HNmMM-derived primary cell lines. Immunofluorescence staining of two HNmMM tissue samples demonstrated expression of PRR and AT_2_R by the SOX2+ CSCs within the TNs and the OCT4+ CSCs within the PTS, with ACE localized to the endothelium of the tumor microvessels within the PTS.

## 1. Introduction

Malignant melanoma (MM) accounts for 62% of all deaths caused by skin cancer [1]. Australia and New Zealand carry the most significant mortality burden [2,3], where the incidence of MM has been steadily increasing over the last 30 years [4]. It is the third most common cancer in New Zealand men and women, with up to 35 new cases per 100,000 population annually [5]. 10–20% of MM affect the head and neck [6]. While the five-year survival rate for MM is around 92% for stages I and II disease, this drops significantly to 5–23% for metastatic MM, with an average overall survival of approximately nine months [7]. This poor survival outcome has been attributed to the presence of cancer stem cells (CSCs) [8,9].

CSCs are pluripotent stem cells with embryonic stem cell (ESC) properties including unlimited proliferative potential and the ability to undergo asymmetric division, giving rise to identical CSCs and differentiated cancer cells [10]. These differentiated cancer cells rapidly proliferate, making up the bulk of the tumor, and are susceptible to radiotherapy and chemotherapy which target rapidly dividing cells. In contrast, CSCs are quiescent, as supported by their local niche [8,11,12,13], which has also been described in MM [14,15]. CSCs have been shown to resist radiotherapy and chemotherapy and are responsible for loco-regional recurrence and distant metastasis [8,9,16,17,18]. We have identified CSC populations in several cancer types, including glioblastoma [19], renal clear cell carcinoma [20], oral cavity squamous cell carcinoma (SCC) of different subsites [21,22,23], primary [24] and metastatic [25] head and neck cutaneous SCC, primary [26] and metastatic [27] colon adenocarcinoma, metastatic MM to the brain [28], and head and neck metastatic MM (HNmMM) [29].

The endocrine RAS cascade regulates blood pressure and fluid and electrolyte homeostasis [30]. In response to low blood pressure, low blood sodium levels, or sympathetic stimulation via activation of β-adrenergic receptors, the juxtaglomerular cells of the kidneys convert pro-renin to the biologically active renin. Alternatively, pro-renin can be activated by binding to pro-renin receptor (PRR) [31]. Renin then cleaves angiotensinogen that is secreted into the bloodstream by the liver, forming angiotensin I (ATI). ATI is further cleaved by angiotensin-converting enzyme (ACE), predominantly found in the endothelial cells of lung capillaries, generating angiotensin II (ATII). ATII is the main effector of the RAS, acting on ATII receptor 1 (AT_1_R) and ATII receptor 2 (AT_2_R) to increase blood pressure via vasoconstriction and the uptake of sodium and water in the kidneys. ACE2 balances the RAS by metabolizing ATII to angiotensin 1–7, which has contrasting effects to ATII [32].

A local (paracrine) RAS has been identified in many tissue types [33] in which it is involved in the regulation of a wide-range of processes including cellular proliferation, angiogenesis, tumorigenesis, and metastasis [34,35,36,37], with an increasing number of studies showing a relationship between RAS dysregulation and carcinogenesis [36,37,38,39,40]. We have previously demonstrated expression of components of the RAS by CSCs in glioblastoma [41], metastatic MM to the brain [42], primary head and neck cutaneous SCC [43], oral cavity SCC of different subsites [44,45,46], and metastatic colon adenocarcinoma [47]. As abnormal RAS expression is involved in cancer development and metastasis, and therapeutic targeting of CSCs by manipulation of the RAS has been proposed [48,49].

This study aimed to investigate the expression of components of the RAS, by co-staining with the ESC markers SOX2 and OCT4, which were used as the surrogate markers for the OCT4+/SOX2+/KLF4+/c-MYC+ CSCs we have recently identified within the tumor nests (TNs) and the peritumoral stroma (PTS) of human HNmMM to regional lymph nodes [29].

## 2. Materials and Methods 

### 2.1. HNmMM Tissue Samples

Tissue samples of HNmMM to the parotid and/or neck nodes from 16 male and four female patients aged 47–103 (median, 74.5) years with a known primary tumor in the head and neck region (Appendix A), included in our previous study [29], were obtained from the Gillies McIndoe Research Institute Tissue Bank. This study was approved by the Central Health and Disabilities Ethics committee (Ref. 12CEN74). Written consent was obtained from all patients. 

### 2.2. HNmMM-Derived Primary Cell Lines

Primary cell lines were derived from fresh HNmMM tissue samples from four of the original cohort of 20 patients. To generate primary cell lines, small tissue pieces from these tumor samples were incubated between layers of Matrigel (cat#354234, Corning, Tewksbury, MA, USA) in a 24-well plate with media containing Dulbecco’s Modified Eagle Medium (DMEM) with Glutamax™ (cat#10569010, Gibco, Rockford, IL, USA) supplemented with 2% penicillin-streptomycin (cat#15140122, Gibco) and 0.2% gentamycin-amphotericin (cat#R01510, Gibco). Once sufficient cell growth was achieved to support transfer to a monolayer culture, cells were extracted by dissolving Matrigel with Dispase (cat#354235, Corning) and placed in an adherent culture flask with media containing DMEM with Glutamax™ supplemented with 10% fetal calf serum (cat#10091148, Gibco), 5% mTeSR™1 Complete Medium (cat#85850, STEMCELL Technologies, Vancouver, BC, Canada), 1% penicillin-streptomycin, and 0.2% gentamycin-amphotericin in a humidified incubator at 37°C and 5% CO₂. Cells were expanded in culture and harvested between passages 4 and 8. 

### 2.3. Histology and Immunohistochemical Staining

Consecutive 4-µm-thick formalin-fixed paraffin-embedded (FFPE) HNmMM tissue samples from 20 patients underwent hematoxylin and eosin (H&E) staining and Melan-A (ready-to-use; cat#PA0233, Leica, Nussloch, Germany) staining to confirm the presence and diagnosis of MM by an anatomical pathologist. S-100 protein (1:200, cat#330M-16, Cell Marque, Rocklin, CA, USA) was used for cases that were negative for Melan-A. The HNmMM FFPE sections then underwent immunohistochemical staining with primary antibodies against renin (1:500; cat#14291-1-AP, Proteintech, Chicago, IL, USA), PRR (1:500; cat#ab40790, Abcam, Cambridge, UK), ACE (1:300; cat#ab11734, Abcam), ACE2 (1:200; cat#MAB933, R&D Systems, Minneapolis, MN, USA), and AT_2_R (1:2000; cat#NBP1-77368, Novus Biologicals, Littleton, CO, USA) using the Leica BOND RX™ Research Auto-strainer (Leica) with 3,3′-diaminobenzidine as the chromogen. 

Normal human tissues used for positive controls were kidney for renin, ACE, ACE2, and AT_2_R and placenta for PRR. Tissue negative controls were salivary gland for renin and AT_2_R, colon for PRR, and skin for ACE and ACE2. Specificity of the secondary antibodies was confirmed on HNmMM sections with the primary antibody replaced with an appropriate isotype matched antibody. We were unable to validate a suitable antibody for AT_1_R and have therefore excluded this marker from protein-level analysis [50,51,52,53].

### 2.4. Reverse Transcription Quantitative Polymerase Chain Reaction

From the original cohort of 20 patients, five snap-frozen HNmMM tissue samples and four HNmMM-derived primary cell lines were analyzed for expression of renin, PRR, ACE, ACE2, and AT_2_R. Approximately 20mg from each HNmMM tissue sample was homogenized using the OMNI tissue homogenizer (OMNI International, Kennesaw, GA, USA) followed by total RNA extraction using the RNeasy Mini kit (cat#74104, Qiagen, Hilden, Germany). Total RNA was extracted from frozen cell pellets of 5x10^5^ viable cells from each of the four HNmMM-derived primary cell lines using the RNeasy Micro kit (cat#74004, Qiagen). An on-column DNase digest (cat#79254, Qiagen) step was included for each extraction. RNA was quantified using the NanoDrop 2000 spectrophotometer (Thermo Fisher Scientific, Waltham, MA, USA). One-step reverse transcription quantitative polymerase chain reaction (RT-qPCR) was run on the Rotor Gene-Q (Qiagen) to assess gene expression using the Rotor-Gene Multiplex RT-PCR kit (cat#204974, Qiagen) and TaqMan Gene Expression Assay primer probes (cat#4331182, Thermo Fisher Scientific). The probes used were renin (Hs00982555_m1), PRR (Hs00997145_m1), ACE (Hs00174179_m1), ACE2 (Hs01085333_m1), AT_1_R (Hs00258938_m1), and AT_2_R (Hs00169126_m1). Gene expression was normalized against the housekeeping genes *GAPDH* (Hs99999905_m1) and *PSMB4* (Hs00160598_m1). Nuclease-free water was added to the no template RNA as a negative control. PC3 cells were used as a positive control for renin, uterine fibroid tissue for PRR, ACE, AT_1_R, AT_2_R, and HepG2 cells for ACE2. Universal human reference RNA (UHR; cat#636690, Takara Bio, Shiga, Japan) was used as a calibrator for the 2^∆∆CT^ (fold-change) analysis. End-point amplification products were run on 2% agarose eGel (cat#G402002, Invitrogen) electrophoresis and imaged on the ChemiDoc MP (Bio-Rad, Hercules, CA, USA) to confirm probe specificity by ensuring the bands observed matched the expected amplicon size of respective target genes. Graphs were generated using GraphPad Prism (v8.0.2, San Diego, CA, USA) and results presented as fold-change relative to UHR. A two-fold change 2^ΔΔCT^ of >2.0 or <0.5 was considered a biologically significant change in expression.

### 2.5. Western Blotting

20 μg of total protein extract from six snap-frozen HNmMM tissue samples and two HNmMM-derived primary cell lines from the original cohort of 20 patients were used for western blotting (WB). Protein was separated by SDS-PAGE with Bolt™ 4–12% Bis-Tris Plus 10 well gels (cat#NW04122BOX, Invitrogen) and transferred from the gel to a PVDF membrane (cat#IB24001, Invitrogen) using the iBlot 2 (Thermo Fisher Scientific). The PVDF membrane was subsequently blocked with iBind™ Flex FD solution (cat#SLF2019, Invitrogen) and probed on the iBind system (Invitrogen). Primary antibodies used were rabbit anti-PRR (1:250; cat#ab40790, Abcam), goat anti-ACE (1:200; cat#sc-12184, Santa Cruz Biotechnology, Dallas, TX, USA), mouse anti-ACE2 (1:500; cat#MAB933, R&D Systems), rabbit anti-AT_2_R (1:500; cat#ab92445, Abcam), and mouse anti-α-tubulin (1:2000; cat#62204, Invitrogen). Secondary antibodies were goat anti-rabbit HRP for PRR (1:1000; cat#ab6721, Abcam), donkey anti-goat HRP for ACE (1:1000; cat#ab97120, Abcam), goat anti-mouse HRP for ACE2 (1:1000; cat#ab6789, Abcam), donkey anti-rabbit HRP for AT_2_R (1:1000; cat#SA1-200, Invitrogen), and donkey anti-mouse Alexa Fluor 488 for α-tubulin (1:1000; cat#A-21202, Invitrogen). Clarity Western enhanced chemiluminescence substrate (cat#1705061, Bio-Rad) in a ChemiDoc MP Imaging System (Bio-Rad) and Image Lab 5.0 software (Bio-Rad) were used for fluorescent detection of the fluorophore used for α-tubulin, and for HRP visualization of membrane bound target RAS proteins. Sample numbers used to blot for ACE2 vary from the other markers due to sample availability. Antibodies for renin failed to produce the expected result with control samples and were therefore not used with tissue samples.

### 2.6. Immunofluorescence Staining

4-µm-thick FFPE sections of two representative HNmMM tissue samples of the original cohort of 20 patients underwent immunofluorescence dual-staining. ACE was co-stained with the endothelial marker ERG (1:200; cat#434R-16, Cell Marque), while PRR and AT_2_R were co-stained with the ESC markers SOX2 (1:100; cat#14-9811-82, Invitrogen, Carlsbad, CA, USA) and OCT4 (1:30; cat#309M-16, Cell Marque). Appropriate secondary antibodies and amplification kits were used for immunofluorescence detection; Alexa Fluor anti-mouse 488 (1:500; cat#A-21202, Invitrogen), Alexa Fluor anti-rabbit 594 (1:500; cat#A-21207, Invitrogen), Alexa Fluor anti-rat 647 (1:500; cat#A-21247, Invitrogen), VectaFluor Excel anti-mouse 488 (ready-to-use; cat#DK-2488, Vector Laboratories, Burlingame, CA, USA), and VectaFluor Excel anti-rabbit 594 (ready-to-use; cat#DK-1594, Vector Laboratories). Primary isotype controls were sections of HNmMM incubated with rabbit antibody (ready-to-use; cat#IR600, Dako, Glostrup, Denmark), mouse antibody (ready-to-use; cat#IR750, Dako), or rat antibody (1:100; cat#14-4321-85, Invitrogen), confirming the specificity of secondary antibodies. Immunofluorescence slides were mounted in Vectashield Hardset mounting medium with 4′,6-diamino-2-phenylindole (cat#H-1500, Vector Laboratories). All antibodies were diluted in BOND primary antibody diluent (cat#AR9352, Leica).

### 2.7. Image Analysis and Capture

Immunohistochemical-stained slides were viewed and imaged using an Olympus BX53 microscope fitted with an Olympus SC100 digital camera (Olympus, Tokyo, Japan), and processed with the cellSens 2.0 Software (Olympus). Immunofluorescence-stained slides were viewed and imaged using an Olympus FV1200 biological confocal laser-scanning microscope (Olympus) and processed with the cellSens Dimension 1.11 software.

## 3. Results

### 3.1. PRR, ACE, and AT_2_R, but Not Renin or ACE2, Were Expressed in HNmMM Tissue Samples

H&E staining confirmed the presence of HNmMM in all 20 tissue samples, organized into TNs with intervening PTS (Figure 1A). Immunohistochemical staining showed no expression of renin (Figure 1B) in 19 samples, with one sample showing weak diffuse staining throughout the tumor (data not shown). PRR (Figure 1C) was mainly expressed in the cytoplasm of the cells within the TNs, and to a lesser extent cells within the PTS. ACE (Figure 1D) was present on the endothelium of the tumor microvessels, mainly located within the PTS. ACE2 (Figure 1E) was not detected in any of the samples. AT_2_R (Figure 1F) was expressed on the nuclear membranes of cells within the PTS, and in the cytoplasm of cells within the TNs. 

Human tissues used for positive controls for immunohistochemical staining showed the expected staining patterns for renin in kidney (Appendix A), PRR on placenta (Appendix A), and ACE (Appendix A), ACE2 (Appendix A), and AT_2_R (Appendix A) in kidney. HNmMM samples with the relevant isotype controls (Appendix A) and tissue negative controls (Appendix A) demonstrated minimal staining.

### 3.2. Transcripts of PRR, ACE, ACE2, and AT_1_R But Not Renin or AT_2_R, Were Expressed in HNmMM Tissue Samples and HNmMM-Derived Primary Cell Lines

RT-qPCR of five HNmMM tissue samples and four HNmMM-derived primary cell lines confirmed expression of PRR, ACE, and ACE2, but not AT_2_R while AT_1_R was detected in four out of five tissue samples and three cell samples, and renin was detected in only one tissue sample (Figure 2A) and one cell line (Figure 2B). There was a slight (biologically insignificant) decrease in PRR expression in the tissue samples (Figure 2A), with no significant change in the cell lines relative to UHR (Figure 2B). ACE expression was downregulated in all five HNmMM tissue samples (Figure 2A) and four HNmMM-derived primary cell lines (Figure 2B), relative to UHR. ACE2 was observed in all tissue samples (Figure 2A) and three of the four cell lines (Figure 2B), with expression downregulated, relative to UHR. AT_1_R expression was detected in all but one of the five tissue sample (Figure 2A), and in three of the four cell lines (Figure 2B). Expression was downregulated relative to UHR in tissue samples, but not in cell line samples. AT_2_R was not detected in any of the tissue samples (Figure 2A) or HNmMM-derived primary cell lines (Figure 2B). Specific amplification of the products was demonstrated by electrophoresis of qPCR products on 2% agarose gels (Appendix A). The expected size amplicons were observed, and no products were observed in the no template control reactions (Appendix A). 

### 3.3. PRR, ACE, and AT_2_R But Not ACE2 Were Heterogeneously Expressed in HNmMM Tissue Samples and HNmMM-Derived Primary Cell Lines

WB was performed to investigate the presence of PRR, ACE, ACE2, and AT_2_R proteins in six HNmMM tissue samples and two HNmMM-derived primary cell lines. PRR (Figure 3A) was present in five out of the six HNmMM tissue samples and both HNmMM-derived primary cell lines at the expected size for the soluble form of PRR at 28k Da [54]. A band representing ACE (Figure 3B) was detected at 190 kDa in three of the six tissue samples, but not in the two cell lines. This may reflect the heterogeneity of these tumors but more likely is due to an abundance of blood vessels in some areas of tissue sections and not the others. AT_2_R (Figure 3C) was present at 45 kDa in four of the six tissue samples but was not detected in either of the two cell lines. ACE2 (Figure 3D) was not detected in any of the tissue samples or cell lines, with an appropriate band present at 110 kDa in the positive control. The sample numbers for ACE2 are different due to availability of samples over the course of the project. α-Tubulin at ~50 kDa confirmed approximately equal total protein loading for all samples probed for PRR, ACE and AT_2_R (Appendix A) and ACE2 (Appendix A). WB for renin was abandoned after multiple antibodies failed to produce a single specific band in positive control tissues.

### 3.4. PRR and AT_2_R Were Localized to CSCs, and ACE to the Endothelium of the Microvessels in HNmMM Tissue Samples

To investigate the localization of components of the RAS in relation to the CSC subpopulations within HNmMM, immunofluorescence dual-staining was performed using antibodies against PRR and AT_2_R with SOX2 or OCT4; and ACE with the endothelial cell marker ERG. SOX2 and OCT4 were chosen as surrogate markers to identify CSCs as they were expressed by CSC subpopulations we have previously identified in HNmMM [29]. 

PRR (Figure 4A, red) showed mainly cytoplasmic expression in the SOX2+ (Figure 4A, yellow) CSCs within the TNs, and to a lesser extent, SOX2 cells within the PTS. PRR (Figure 4B, red) also showed cytoplasmic expression in cells within the PTS which expressed OCT4 (Figure 4B, green). Cytoplasmic expression of ACE (Figure 4C, green) was demonstrated on the endothelium of the tumor microvessels which showed nuclear staining of ERG (Figure 4C, red). AT_2_R (Figure 4D, red) showed membranous staining of SOX2+ (Figure 4D, yellow) CSCs within the TNs and some cells within the PTS. AT_2_R (Figure 4E, red) was also expressed by the OCT4+ (Figure 4E, green) CSCs within the PTS. Renin and ACE2 were not stained using immunofluorescence given that they were negative on immunohistochemical staining. Figure insets have been provided to show enlarged views of the corresponding images. Split immunofluorescence images are provided in Appendix A. No staining was present on the negative controls (Appendix A).

## 4. Discussion

This study investigated whether components of the RAS—renin, PRR, ACE, ACE2, AT_1_R, and AT_2_R—were locally expressed by the CSC subpopulations that we have previously identified in HNmMM [29]. Five of these six components of the RAS were shown to be present in the HNmMM tissue samples by at least one of the methods used in this study. PRR and ACE were found using all techniques, ACE2 and AT_1_R using RT-qPCR only, and AT_2_R using IHC and WB. These differences could be explained by the heterogeneity of these tumors. 

ACE was detected by WB in tissue samples, but not in HNmMM-derived primary cell lines. This can be explained by the potential loss of the vascular endothelial cell population during culturing. For ACE2, significant downregulation in the order of 200x, or lack of translation may explain why we found mRNA but no protein expression. The probe for AT_2_R that was used in RT-qPCR analysis does not detect a predicted splice variant, which may explain why protein expression was found in the absence of mRNA. In addition, the interpretation of biologically significant changes in mRNA expression may be altered when compared to UHR, in the absence of a specific similar normal tissue control. Unfortunately, we were unable to investigate the presence of AT_1_R at the protein level due to the lack of suitable antibodies [50,51,52,53]. The lack of AT_2_R expression in the HNmMM-derived primary cell lines but its presence in tissue samples suggests its expression may rely on extracellular signals and thus is not expressed in cell lines which lack an ECM. Alternatively, the cells derived from the culture of HNmMM tissue samples may be exclusively those not expressing AT_2_R.

Immunofluorescence staining demonstrated expression of AT_2_R by the CSCs that we have previously identified, using SOX2 and OCT4 as representative markers for the CSC subpopulations we have identified in HNmMM [29]. This is consistent with previous research showing expression of components of the RAS by the CSCs in metastatic MM to the brain [42]. SOX2 and OCT4, along with other induced pluripotent stem cell markers KLF4, NANOG, and c-MYC, regulate stemness in CSCs [55] and have been used them to characterize CSCs in other cancer types [19,20,21,22,23,24,25,26,27,28,29]. Specifically, in HNmMM, two OCT4+/SOX2+/KLF4+/c-MYC+ subpopulations were identified, one within the TNs, and one within the PTS [29]. SOX2 is known to regulate MM CSC self-renewal and tumorigenicity [56], as well as contributing to tumor invasion [57]. Overexpression and transmembrane delivery of OCT4 in MM has been demonstrated to drive dedifferentiation to CSC-like cells, with associated resistance to chemotherapy and hypoxia, and increased tumorigenesis [58]. Its expression indicates that the HNmMM cells have acquired a stem-cell-like phenotype, indicating that a subset of tumor cells within HNmMM have acquired stem-cell-like qualities and utilize the RAS pathway. The expression of components of the RAS in metastatic MM at different sites may indicate a role for the RAS in the acquisition of a malignant phenotype.

The role of renin in cancer has not been well studied, although it has been found to be upregulated in various cancer types [36], and attenuation of renin has been shown to reduce cell growth and induce apoptosis in glioblastoma [59]. However, we did not detect renin in our HNmMM samples. This does not necessarily rule out the potential of local generation of ATI and ATII, as there are bypass loops of the RAS [34,35,36]. Cathepsin D constitutes one such bypass loop and is known to cleave angiotensinogen to ATI [60,61]. Enzymatically active cathepsin D has been demonstrated in MM [62], localized to malignant melanocytes [63]. Moreover, increased cathepsin D expression has been noted in metastatic MM, compared to primary MM [64]. In addition, cathepsin D has been identified as a potential biomarker in uveal melanoma using an unbiased proteomics approach [65].

PRR has been shown to contribute to carcinogenesis via a variety of pathways, including Wnt/β-catenin signaling, MAPK/ERK and PI3K/AKT/mTOR pathways, V-ATPase, and the RAS itself [35,66]. 

There are multiple processes through which ACE affects carcinogenesis, including cell proliferation and differentiation, angiogenesis, and immune sensitization [67]. Multiple in vitro and in vivo studies have demonstrated a reduction in tumor growth [68,69,70,71,72,73,74], angiogenesis [68,69,70,71], and metastasis [71,72,73] with administration of various ACE inhibitors. Conversely, overexpression of ACE in macrophages in a mouse model shows increased resistance to MM, mediated through CD8+ T cells, although it is interesting to note that this persists despite AT_1_R receptor blockade, with the knock-out of angiotensinogen strengthening this effect [40]. The role of ACE in angiogenesis may explain why it is localized to the endothelium of the tumor microvessels, as demonstrated in this study.

ACE2, acting through the ACE2/AT(1–7)/Mas axis, has been shown to suppress cancer through a variety of mechanisms—namely, cell proliferation, invasion, migration, angiogenesis, and epithelial-to-mesenchymal transition [75]; the absence of ACE2 is therefore significant. 

Recent work has demonstrated that AT_1_R expression inhibits proliferation in human melanoma cell lines, with inhibition of the receptor promoting proliferation [76]. This is in direct contrast to the normal observed roles of the classical endocrine RAS and the mechanisms of action in other cancers and murine melanoma models in which upregulated AT_1_R expression promotes tumor growth [37,38,39]. Unfortunately, without the availability of specific antibodies we were unable to directly demonstrate protein expression of AT_1_R and its localization to the CSCs in HNmMM tissues. The absence of ACE2, however, suggest that there may be increased signaling through AT_1_R. 

AT_2_R counteracts the detrimental effects of AT_1_R in cancer [77] and reduces pancreatic carcinoma growth in a murine model [78]. In addition, signaling through AT_2_R induces apoptosis of cancer cell lines independently of ATII [79]. This has important implications when considering how modulation of the RAS may be utilized as a treatment strategy. 

Epidemiological studies have demonstrated conflicting effects of RAS inhibition in cancer [80], with studies showing no significant difference with ACE inhibitor or angiotensin receptor blocker use [80], although a more recent clinical trial investigating the neoadjuvant use of losartan demonstrates downstaging in pancreatic cancer [81]. 

In this study, we have shown local expression of components of the RAS; PRR, ACE, and AT_2_R, but not ACE2 in all HNmMM tissues samples, while renin was detected in only one of the samples. PRR was localized to CSCs within the TNs, while AT_2_R was expressed in the CSCs within both the TNs and the PTS. ACE is localized to the endothelium of the tumor microvessels within the PTS. Further work with larger sample size and including functional studies may provide greater insights into the biology and novel therapeutic targeting of CSCs in HNmMM the treatment of this aggressive cancer by manipulation of the paracrine RAS. 

## 5. Patents

There are no patents resulting from the work reported in this manuscript.

## Figures and Tables

**Figure 1 life-10-00268-f001:**
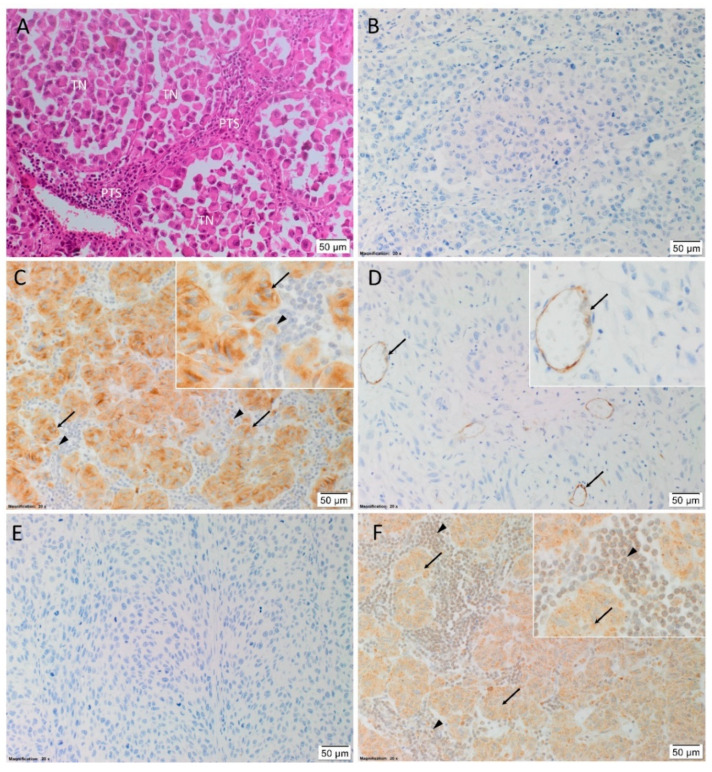
Representative hematoxylin and eosin and immunohistochemical stained images showing the tumor architecture and the localization of components of the renin-angiotensin system (RAS) in head and neck metastatic malignant melanoma (HNmMM) tissue samples. HNmMM tissue demonstrating the tumor nests (TNs) surrounded by peritumoral stroma (PTS) (**A**). There was no expression of renin ((**B**), brown). Pro-renin receptor (PRR) ((**C**), brown) was present in cells within the TNs (*arrows*) and the PTS (*arrowheads*). ACE ((**D**), brown) was localized to the endothelium of the tumor microvessels within the PTS (*arrowheads*). ACE2 ((**E**), brown) was not expressed. Angiotensin II receptor 2 (AT_2_R) ((**F**), brown) was present in cells within the TNs (*arrows*) and the PTS (*arrowheads*). Nuclei were counterstained with hematoxylin ((**A**–**F**), blue). Original magnification 200× *n* = 20. The inserts show enlarged views of the corresponding images.

**Figure 2 life-10-00268-f002:**
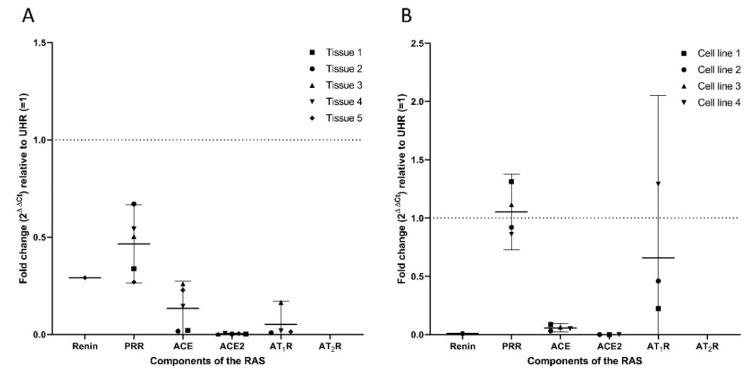
Reverse transcription quantitative polymerase chain reaction (RT-qPCR) analysis of five head and neck metastatic malignant melanoma (HNmMM) tissue samples and four HNmMM-derived primary cell lines demonstrating expression of PRR, ACE, ACE2, and AT_1_R, but not renin or AT_2_R. HNmMM tissue (**A**), and HNmMM-derived primary cell lines (**B**) were analyzed for expression of transcripts of components of the renin-angiotensin system (RAS) with one-step RT-qPCR. Renin was detected in only one tissue sample and one cell line, with reduced expression relative to universal human reference RNA (UHR). PRR was detected in all samples, with reduced expression in tissue samples, and no change in cell lines relative to UHR. ACE was also detected in all tissue samples and cell lines, with expression downregulated, relative to UHR. AT_1_R was detected in four tissue samples, and three cell lines. AT_2_R was not detected in any of the tissue samples or cell lines. Expression was normalized to the reference genes *GAPDH* and *PSMB4*, error bars represent 95% confidence intervals of the mean.

**Figure 3 life-10-00268-f003:**
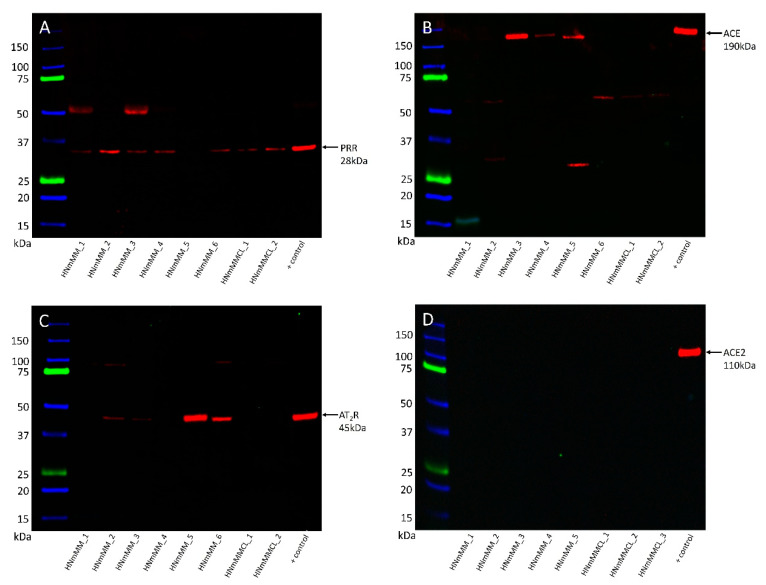
Representative western blot images of head and neck metastatic malignant melanoma (HNmMM) tissue samples and HNmMM-derived primary cell line samples show varying expression levels of the components of the renin-angiotensin system. PRR ((**A**), ~28 kDa) was present in five out of six tissue samples and both HNmMM-derived primary cell lines at the expected size for the soluble form. ACE ((**B**), ~190 kDa) was only seen in three of the tissue samples. AT_2_R ((**C**), ~45 kDa) was present in four of the tissue samples, but none of the cell lines. ACE2 ((**D**), ~110 kDa) was not detected in any of the five tissue samples or three cell lines.

**Figure 4 life-10-00268-f004:**
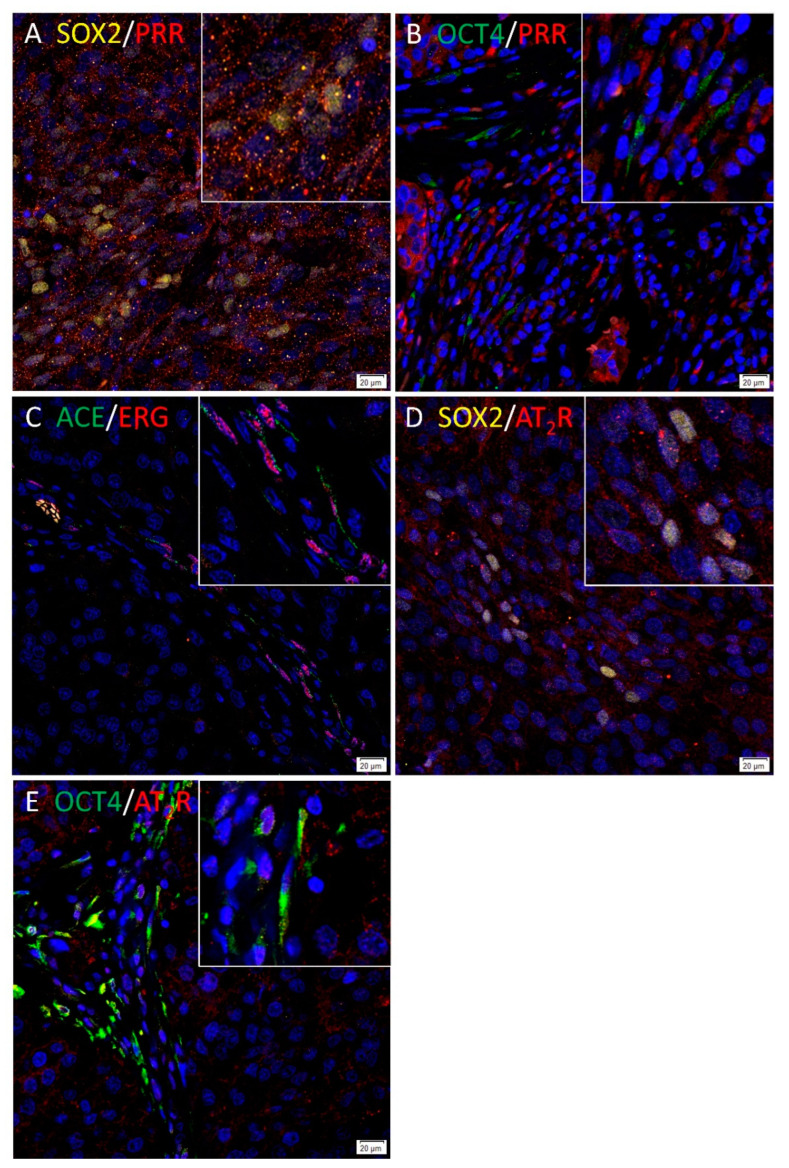
PRR and AT_2_R were expressed by the cancer stem cells (CSCs) in head and neck metastatic malignant melanoma (HNmMM), with ACE localizing to the endothelium of microvessels. PRR ((**A**), red) was expressed by the SOX2+ ((**A**), yellow) CSCs within the tumor nests (TNs) and some cells within the peritumoral stroma (PTS). PRR ((**B**), red) was also expressed by cells within the TNs and the PTS, some of which expressed OCT4+ ((**B**), green). ACE ((**C**), green) was localized to the ERG+ ((**C**), red) endothelium of the tumor microvessels within the PTS. AT_2_R ((**D**), red), showed membranous expression of SOX2+ ((**D**), yellow) CSCs within the TNs and some cells within the PTS. AT_2_R ((**E**), red) was localized to OCT4+ ((**E**), green) CSCs within the PTS. All slides were counterstained with 4′,6-diamidino-2-phenylindole ((**A**–**E**), blue). Original magnification 400×; *n* = 2. The inserts show enlarged views of the corresponding images.

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
