# Peer review of "Cancer Stem Cells in Head and Neck Metastatic Malignant Melanoma Express Components of the Renin-Angiotensin System"

_life, 2020, doi:10.3390/life10110268_

Round 1

Reviewer 1 Report

This manuscript describes the occurrence of compounds of the RAS in CSCs of metastatic melonoma in human patients, suggesting a functional role of the RAS or of compounds of the RAS in malignant melanoma.

The manuscript is clearly written and easy to understand, the experiments are well executed and in particular the rigid use of positive and negative controls is very well done.

As the findings may be relevant for both understanding the malignancy of melanoma and possibly therapy, I find this a relevant publication and I advise minor revision only.

  1. My major comment is the number of patients. The original number of patients is 20. Only 5 snap-frozen samples and 4 primary cell lines were used for RT-qPCR, and 6 snap-frozen samples and 2 primary cell lines were used for WBwhereas only 2 samples were used for IF staining. There is no reasoning why only a limited number of patient samples were used for the different analyses. On what basis was the selection made? Was there any prevention measure to avoid bias? In the Legends it is stated that representative images are shown. On what basis it was kbnown that these are representative?
  2. In line 18 of the Abstract information on the samples should be given, that is now done in lines 22-23. That is confusing.
  3. Lines 41-50. It is correctly stated that CSCs have unlimited proliferative potential and can undergo asymmetric division, whereas the differentiated derived cancer cells are rapidly proliferating and thus susceptible to therapy. It is not clearly stated that the CSCs are quiescent and thus less sensitive to therapy and in my opinion this is a crucial difference between CSCs in their niche and the differentiated proliferating cancer cells as occur in normal stem cells versus non-stem cells such as in the bone marrow (Hira et al (2020) J Histochem Cytochem 68:33-57 and (2020) Biology 9:31. Please, comment upon these aspects of CSCs in niches.
  4. In lines 71-73 it is stated that CSCs that RAS was investigated in relation to positivety for OCT4, SOX2, KLF4, and MYC, but the authors tested only for positivity of SOX2 and OKT4.
  5. Lines 280-285. Here it should be discussed whether the endocrine or the paracrine expression of RAS compounds are involved in the malignant cells, because it is stated in the Intro (lines 62-65) that the paracrine RAS is linked with cell proliferation and malignancy. Please, comment.

When these comments have been addressed properly in a revised manuscript, it is acceptable for publication, in my opinion.

Reviewer 2 Report

This is a well-written report showing expression of several CSCs markers in HNmMM. Results are technically valid. I have only minor suggestions.

  1. Please provide statistical analysis for RT-PCR data.
  2. The study would benefit From adding data available in public databases on expression of these genes in other studies.
